# Towards Better Representations for Multi-Label Text Classification with Multi-granularity Information

**Fangfang Li[1], Puzhen Su[1], Junwen Duan[1]***, **Weidong Xiao[2]**

[1]School of Computer Science and Engineering, Central South University
[2]Science and Technology on Information Systems Engineering Laboratory,
National University of Defense Technology
{lifangfang,214711091,jwduan}@csu.edu.cn, wdxiao@nudt.edu.cn

## Abstract

Multi-label text classification (MLTC) aims to assign multiple labels to a given text. Previous works have focused on text representation learning and label correlations modeling using pre-trained language models (PLMs). However, studies have shown that PLMs generate word frequency-oriented text representations, causing texts with different labels to be closely distributed in a narrow region, which is difficult to classify. To address this, we present a novel framework, **CL**(Contrastive Learning)-**MIL** (Multi-granularity Information Learning), to refine the text representation for MLTC task. We first use contrastive learning to generate uniform initial text representation and incorporate label frequency implicitly. Then, we design a multi-task learning module to integrate multi-granularity (diverse text-labels correlations, label-label relations and label frequency) information into text representations, enhancing their discriminative ability. Experimental results demonstrate the complementarity of the modules in CL-MIL, improving the quality of text representations, and yielding stable and competitive improvements for MLTC.

## 1 Introduction

Multi-Label Text Classification (MLTC) is a fundamental task in the natural language processing (NLP) field which refers to assigning multiple relevant labels to each text. MLTC has been widely applied to various real-world scenarios like recommendation system (Guo et al., 2016), sentiment analysis (Wang et al., 2016) and information retrieval (Yang and Gopal, 2012) and others (Jain et al., 2016; Papanikolaou et al., 2014; Zhang et al., 2014). In MLTC, there are diverse text-labels correlations. As shown in Figure 1, labels $L_1, L_2$ are directly correlated to specific type of keywords and phrases (fine-grained) $T_2, T_4$, whereas latent correlations are determined by multiple information

---
*Corresponding author

(coarse-grained). For example, $L_4$ is jointly correlated with $T_2, T_{n-2}$ and $T_n$.

Previous methods solve MLTC task in diverse perspectives. Among them, some methods focus on fine-grained (local semantic features) (Kurata et al., 2016) and coarse-grained (global semantic features) textual information (Liu et al., 2016), respectively. Besides, (Wang et al., 2018; Xiao et al., 2019) utilize joint embedding and label-wise attention to enhance the text-labels correlations, and other methods learn the correlations between labels (Nam et al., 2017; Yang et al., 2018). These methods separately utilize information from different granularities to improve the classification performance. However, learning information on single granularity is insufficient to derive effective text representations compared to keeping a proper balance of multi-granularity information.

Recently, PLMs (Devlin et al., 2019; Peters et al., 2018) have been proved powerful to generate high-quality semantic representations. Competitive and strong PLMs based methods (Zhang et al., 2021; Su et al., 2022) are proposed to explore labels correlations and knowledge between samples. Yet, (Gao et al., 2019; Zhou et al., 2022; Ethayarajh, 2019a) reveal that PLMs appear anisotropy problem after fine-tuned, leading text representations collapse into a narrow cone space. Texts consisting of high-frequency words are gathered closely, even correlated to extremely low-frequency labels. When samples with low-frequency labels concentrate densely, it will become more difficult to classify and also aggravate the long-tail problem (Menon et al., 2021) due to the lack of samples.

To address these, we propose a multi-granularity information enhancement framework to improve the quality of representations and relieve the anisotropy problem. We first perform contrastive learning to implicitly introduce label frequency into the initial representations and refine the uniformity of it. It only focuses on *target samples* (correlated

to low-frequency labels), and push away *source samples* that share low label co-occurrence with labels in *target sample* according to the label distribution. Moreover, we adopt the *Multi-granularity Information Learning* (MIL) module via multi-task learning to enhance the expressiveness of representations. Specifically, MIL module explores direct and latent text-labels correlations through *Dual-grained Interactive Learning* (DIL) task and models relations (*strong relate*, *weak relate* and *contradict*) between labels by *Constraint Label Relations Learning* (CLRL) task.

Both tasks share the same encoder that jointly embeds text and labels to directly introduce label frequency into representations, which provides MIL an access to transform the orientation of anisotropy problem. In DIL, we use two different interactive learning tasks to obtain both direct and latent text-labels correlations. In CLRL, we utilize label frequency and labels co-occurrence to explore the constraint relations, since the correlations between labels exists not only in single instance, predicting the label correlations from one instance may contradict with the the others.

We illustrate that our method outperforms a series of competitive baseline on AAPD and RCV1-V2 datasets by experimental results. We summarize our contributions as follows:

1. We propose a multi-granularity information enhancement framework that explores diverse text-label correlations and label-label relations to improve the quality of text representations.

2. To alleviate the anisotropy problem, we introduce label frequency into text representations using contrastive learning and the MIL module.

3. Experimental results on two MLTC datasets demonstrate the effectiveness and competitiveness of our method.

## 2 Related Work

**Multi-label Text Classification** In MLTC, early works like CNN (Kim, 2014; Kurata et al., 2016; Poria et al., 2017) and RNN (Liu et al., 2016) were proposed to learn fine-grained and coarse-grained text information. Besides, Wang et al. (2018); Xiao et al. (2019); Chen et al. (2021) adopted joint embedding and text-labels fusion strategy to explore the correlations between text and labels. To capture

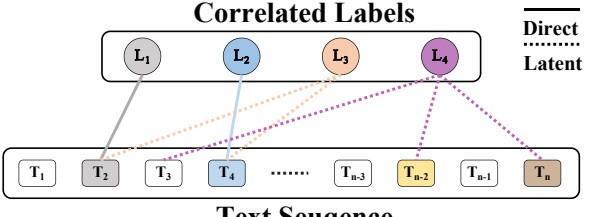

Figure 1: Illustration of different text-labels correlations. The direct correlations are determined by one specific type of key-words, while the latent correlations require a mix of multiple features.

label correlations, (Tsoumakas and Katakis, 2007) and (Read et al., 2011) used unique label combinations and classification chain. Seq2seq-based methods (Nam et al., 2017; Yang et al., 2018) transform MLTC into label sequence generation tasks. These methods tackle MLTC task via different granularities information, which strikes us the feasibility of balancing multi-granularity information to improve the expressiveness of text representations.

**Anisotropy problem** Recently, pre-trained language models (PLMs) (Devlin et al., 2019; Liu et al., 2019) have become paradigms for various NLP tasks. PLMs based methods (Adhikari et al., 2019; Zhang et al., 2021; Su et al., 2022) further achieve remarkable performance on MLTC. However, Ethayarajh (2019b); Su et al. (2021) have demonstrated the anisotropy problem limits the expressiveness of text representations. The representations generated by PLMs are constrained within a narrow cone-shaped space after fine-tuned. Li et al. (2020) reveals that texts with high-frequency words concentrate densely in the narrowest part of the cone, while texts with low-frequency words distribute sparsely. The anisotropy problem in MLTC, owing to high word frequency, leads samples with low-frequency labels to be closely gathered with other samples of different label frequencies, which can corrupt semantic information of representation and hinder the classification performance of MLTC. Therefore, we use contrastive learning to improve the uniformity of representations and introduce label frequency information into representations to alleviate the anisotropy problem.

## 3 Proposed Method

Our framework, illustrated in Figure 2, consists of two modules: a *contrastive learning* (CL) module and a *multi-granularity information learning* (MIL) module. The CL module refines initial rep-

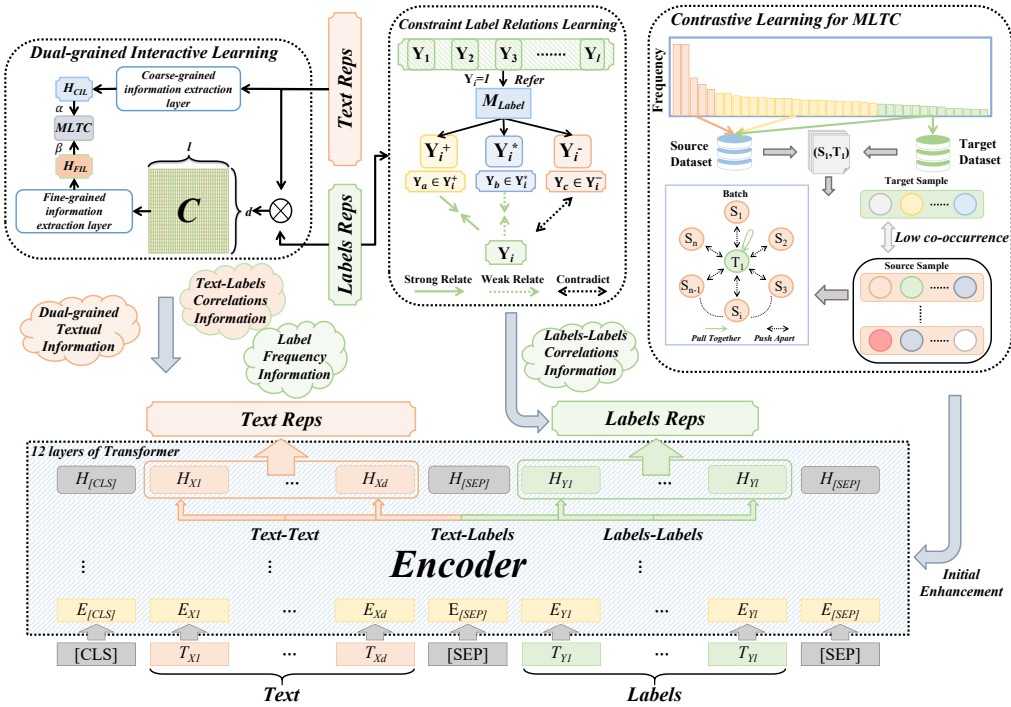

Figure 2: The framework of our proposed approach.

resentations for the MIL module, which introduces multi-granularity information to enhance overall expressive ability. Besides, label frequency is also introduced by both CL and MIL module to alleviate the anisotropy problem.

## 3.1 Problem Formulation

Given a MLTC training set $D = \{(x_i, y_i) | 1 \leq i \leq N\}$ consisting of $N$ samples, each $x_i$ is a text and corresponding to a label sequence $Y_i = \{y_1, y_2, \cdots, y_l | y_k \in \{0, 1\}\}$ consisting of $l$ labels. The aim of MLTC is to learn a mapping function from the text $x_i$ to the corresponding label set $Y_i$.

## 3.2 Contrastive Learning for MLTC

We propose a contrastive learning (CL) objective for the MLTC task to enhance the effectiveness of the MIL module during the fine-tuning stage. This involves providing more uniform and effective initial representations and implicitly introducing label frequency information into it. Besides, considering randomly sampled negatives may bias the representations of samples with similar labels. Our CL objective is based on the labels co-occurrence and mainly focuses on low-frequency labels.

Initially, we delineate two CL datasets: the target dataset, comprising samples associated with low-frequency labels, and the source dataset, encompassing the entirety of the training data. Sub-

sequently, we generate positive $(t_i, t_i^+)$ and negative $(t_i, s_j)$ pairs for CL. To create a positive pair, we select a sample $t_i$ from the target dataset and apply the *dropout* function to obtain $t_i^+$. For negative pairs, we randomly select samples $s_j$ from the source dataset, ensuring that $t_i$ and $s_j$ share negligible or no label co-occurrence. In the specific scenario of a CL data mini-batch, where the batch size is denoted as $N + 1$, our method yields $N$ negative pairs and a single positive pair. The ultimate objective of CL training is articulated as follows:

$$L_{CL} = -\log \frac{e^{sim(\mathbf{t}_i, \mathbf{t}_i^+)/\tau}}{\sum_{j=1}^{N} e^{sim(\mathbf{t}_i, \mathbf{t}_i^+)/\tau} + e^{sim(\mathbf{t}_i, \mathbf{s}_j)/\tau}}$$

(1)

where $sim(\alpha, \beta)$ denotes the cosine similarity, calculated as $\frac{\alpha^\top \beta}{||\alpha|| \cdot ||\beta||}$, $t_i^+$ represents the augmented representation derived from applying *dropout* to $t_i$, and $\tau = 20$ constitutes the temperature hyperparameter.

## 3.3 Multi-Granularity Information Learning (MIL)

The primary objective of the MIL module is to facilitate the learning of multi-granularity information, thereby enriching the representations from diverse perspectives. To address the anisotropy problems and to explore diverse text-label corre-

lations, we propose a *dual-granularity interactive learning* (DIL) task. Concurrently, a *constraint label relations learning* (CLRL) task is proposed to model the relations between labels and deepen text-labels interactive learning. CLRL achieves auxiliary augmentation by optimizing label representations and implicitly passing label relations to the DIL task.

### 3.3.1 Dual-granularity Interactive Learning

In DIL, we design two different granularities interactive leaning tasks to capture the varied text-labels correlations. *Fine-grained interactive learning* (FIL) task achieves interaction based on the output representations of the encoder, and *coarse-grained interactive learning* (CIL) task interacts text with labels through 12 layers of transformer encoder.

**Fine-grained Interactive Leaning** In instances where direct correlations between text and labels are assessed, such as in the **AAPD** dataset, the label *computer vision (CV)* is commonly correlated to specific keyphrases (*image, vision and pixel*) in the given text. To address this, the FIL task interacts text with labels at the token-level to perform fine-grained interactive learning.

We first normalize both the word-level text representation and labels representation by L2 normalization:

$$\widetilde{\mathcal{H}}_{Word} = \ell(\mathcal{H}_{Word}), \widetilde{\mathcal{H}}_{Label} = \ell(\mathcal{H}_{Label}) \quad (2)$$

$$\ell([x_1, \cdots, x_n]) = [\frac{x_1}{\sqrt{\Sigma_{i=1}^h x_{1i}^2}}, \cdots, \frac{x_n}{\sqrt{\Sigma_{i=1}^h x_{ni}^2}}] \quad (3)$$

where $\mathcal{H}_{Word} \in \mathbb{R}^{d \times h}$, $\mathcal{H}_{Label} \in \mathbb{R}^{l \times h}$. Then, we transpose $\widetilde{\mathcal{H}}_{Label}$ and interact with $\widetilde{\mathcal{H}}_{Word}$ by dot product to get *text-labels* correlations matrix $\mathcal{C}$:

$$\mathcal{C} = \widetilde{\mathcal{H}}_{Word} \cdot \widetilde{\mathcal{H}}_{Label}^T \quad (4)$$

where $\mathcal{C} \in \mathbb{R}^{d \times l}$, for $C_{i,j} \in \mathcal{C}$ indicates the interaction between $\mathcal{H}_{x_i}$ and $\mathcal{H}_{Label_j}$. In order to learn the correlations more explicitly, we use *CNN* with *Max-Pooling* under the *ReLU* activation, and obtain correlations score $S \in \mathbb{R}^{d \times 1}$ by *tanh* function. The final representation $\mathcal{H}_{FIL}$ can be calculated by:

$$\mathcal{H}_{FIL} = \mathcal{G}\text{lobal}\mathcal{M}\text{ax}\mathcal{P}\text{ooling}(S \times \widetilde{\mathcal{H}}_{Word}) \quad (5)$$

**Coarse-grained Interactive Leaning** Given that latent correlations between text and labels are influenced by multiple discrete text segments, a global perspective in interactive information exploring is imperative. For example, in **AAPD** dataset, label *machine learning (ML)* is typically correlated to no specific key-phrases, thus the classification of *ML* requires comprehensively integration of the global semantic feature.

In this task, we extract and learn the representation $\mathcal{H}_{(0)} = \{\mathcal{H}_{x_1}, \mathcal{H}_{x_2}, \cdots, \mathcal{H}_{x_d}\}$ through coarse-grained information extraction layer, which is consisted of 15 layers of *Dilated Gated CNN* (DGCNN). Within the DGCNN architecture, the dilated CNN acts to expand the receptive field of the overall structure, thereby enabling to capture more extensive information. Concurrently, the gated framework facilitates transmitting the coarse-grained textual label interactions across multiple channels.

The first layer is calculated by:

$$\mathcal{H}_{(1)} = \mathcal{H}_{(0)} \otimes (1 - \sigma) + \mathcal{C}onv1\mathcal{D}_1(\mathcal{H}_{(0)}) \otimes \sigma$$
$$\sigma = sigmoid(\mathcal{C}onv1\mathcal{D}_2(\mathcal{H}_{(0)}))$$
$$(6)$$

To improve the effectiveness of classifying representations, we use *CNN* and *Multi-head Attention* with a residual structure, and generate the final representation $\mathcal{H}_{CIL}$ by performing *GlobalMaxPooling*:

$$\mathcal{H}_{temp} = \mathcal{M}ulti\mathcal{H}ead\mathcal{A}ttention(\mathcal{H}_{(15)})$$
$$\mathcal{H}_{CIL}^* = \mathcal{C}onv1\mathcal{D}([\mathcal{H}_{temp}, \mathcal{H}_{(15)}]) \quad (7)$$
$$\mathcal{H}_{CIL} = \mathcal{G}lobal\mathcal{M}ax\mathcal{P}ooling(\mathcal{H}_{CIL}^*)$$

**Multi-Label Prediction** For both the $\mathcal{H}_{CIL}$ and $\mathcal{H}_{FIL}$, we take a fully connected layer with *sigmoid* activation as the multi-label classifier, and use Binary Cross Entropy as the loss function:

$$\mathcal{L}_{BCE} = -\sum_{i=1}^{N} \sum_{j=1}^{l} y_{ij} \log(\hat{y}_{ij}) + (1 - y_{ij}) \log(1 - \hat{y}_{ij}) \quad (8)$$

where N is the number of training samples, and l is the length of labels sequence. $y_{ij} \in \{0, 1\}$ and $\hat{y}_{ij} \in [0, 1]$ are the true value and the predicted value of the $y_j$, $y_j \in Y_i = \{y_0, y_1, \cdots, y_l\}$.

### 3.3.2 Constraint Label Relations Learning

The objective of CLRL is to assist the DIL task to perform MLTC by learning similar and constraint relations between labels. As the text and labels are embedded in the same semantic space, the label frequency information is integrated into the DIL task through optimizing label relations. Introducing label frequency thus alleviates the anisotropy problem caused by word frequency bias.

| Label Set | Definition |
|---|---|
| $Y_i^+$ | For any input sample $(D_i, Y_i)$, $y_j \in Y_i = \{y_1, \cdots, y_l\}$ and $y_j = 1$. |
| $Y_i^*$ | For any input sample $(D_i, Y_i)$, $y_j \in Y_i = \{y_1, \cdots, y_l\}$ and $y_j = 0$, $\exists\, y_k \in Y_i^+ \implies \overline{\mathcal{M}}_{y_k} < \mathcal{M}_{y_j, y_k}$, $y_j \notin Y_k^{lowfreq}$. |
| $Y_i^-$ | For any input sample $(D_i, Y_i)$, $y_j \in Y_i = \{y_1, \cdots, y_l\}$ and $y_j = 0$, $\forall\, y_k \in Y_i^+ \implies \overline{\mathcal{M}}_{y_k} > \mathcal{M}_{y_j, y_k}$, $y_j \in Y_k^{lowfreq}$. |

Table 1: Definition of constraint label set

In CLRL, we first analyze the distribution of the training labels set and then count the frequency of co-occurrence between each single label and the rest labels. For a given labels set $Y = \{y_1, y_2, \cdots, y_l\}$, we calculate the label relations matrix $\mathcal{M}_{Label}$ by:

$$\mathcal{M}_{Label} = \sum_{i=1}^{l} \sum_{j=1}^{l} Count(y_i, y_j)$$
$$= \begin{pmatrix} \mathcal{M}_{1,1} & \mathcal{M}_{1,2} & \cdots & \mathcal{M}_{1,l} \\ \mathcal{M}_{2,1} & \mathcal{M}_{2,2} & \cdots & \mathcal{M}_{2,l} \\ \vdots & \vdots & \ddots & \vdots \\ \mathcal{M}_{l,1} & \mathcal{M}_{l,2} & \cdots & \mathcal{M}_{l,l} \end{pmatrix} \quad (9)$$

where $\mathcal{M}_{i,j}$ indicates the frequency of the co-occurrence between label $y_i$ and label $y_j$ in the training dataset.

We then calculate the average frequency co-occurrence of each label by:

$$\overline{\mathcal{M}}_i = \frac{1}{l-1} \sum_{j=1, j \neq i}^{l} \mathcal{M}_{i,j} \quad (10)$$

and then obtain the low-frequency labels sequence $Y_i^{lowfreq} = \{y_a, \cdots, y_b\}$ according to $\overline{\mathcal{M}}_i$, where $\mathcal{M}_{i,a}, \mathcal{M}_{i,b} \leq \overline{\mathcal{M}}_i$.

In light of the foregoing, we randomly select one text-correlated label, denoted as $Y_i$. Subsequent to this, we partition the remaining labels into three distinct subsets: $Y_i^+$, $Y_i^*$, and $Y_i^-$, based on the relations of labels and the low-frequency labels sequence. The definition of these label subsets can be found in Tab 1.

To elucidate further, we first randomly select $y_i$ from the text correlated labels as matching label A, and partition the input labels into three constraint label sets according to matching label A. Then, we randomly select one label $y_j$ from the rest of the labels as matching label B. We obtain the relations embedding $\mathcal{H}_{relations}$ to perform the CLRL by:

$$\mathcal{H}_{relations} = (\mathcal{H}_A, \mathcal{H}_B, |\mathcal{H}_A - \mathcal{H}_B|) \quad (11)$$

Herein, $\mathcal{H}_A$ and $\mathcal{H}_B$ epitomize the representations of matching label A and B, respectively. To explore the relations between labels, we utilize a fully connected layer with a *softmax* activation as the label relations classifier. The adopted loss function is the Categorical Cross Entropy, which can be articulated as:

$$\mathcal{L}_{CCE} = -\sum_{k=1}^{c} y_k \log(\hat{y}_k) \quad (12)$$

where $c$ is the category of the CLRL label, whilst $y_k$ and $\hat{y}_k$ signify the actual and predicted values of category $k$ within the CLRL label, respectively.

In the scenario where Matching Label B aligns with $Y_i^+$, $Y_i^*$ or $Y_i^-$, the CLRL labels are respectively designated as directly related [1,0,0], indirectly related [0,1,0], or irrelevant [0,0,1].

### 3.3.3 Training Objectives

The loss functions for the multi-label text classifier (FIL & CIL) are defined as $\mathcal{L}_{FIL}$ and $\mathcal{L}_{CIL}$, utilizing Binary Cross Entropy $\mathcal{L}_{BCE}$. The loss function for the label relations classifier is denoted as $\mathcal{L}_{CLRL}$, employing Categorical Cross Entropy $\mathcal{L}_{CCE}$. The total loss of MIL can be calculate by:

$$\mathcal{L}_{MIL} = \alpha \mathcal{L}_{CIL} + \beta \mathcal{L}_{FIL} + \lambda \mathcal{L}_{CLRL} \quad (13)$$

where both $\alpha$ and $\beta$ are hyper-parameters in (0,2) and $\lambda$ is hyper-parameter in (0,1) we set $\alpha = 1$, $\beta = 1$, $\lambda = 0.35$ as the final setting. In addition, the final predictions of DIL are the summed by the output of FIL and CIL weighted respectively by $\gamma$ and $1 - \gamma$ during inference stage, where we set $\gamma = 0.5$.

## 4 Experimental Settings

In this section, we evaluate the main experimental results of the baseline models and our method on two datasets. Implementation details of our method can be found in Appendix A.

### 4.1 Datasets

To validate our method, we conduct experiments on two benchmark datasets **AAPD** (Yang et al., 2018) and **RCV1-V2** (Lewis et al., 2004). Tab 2 shows statistics of both datasets.

### 4.2 Evaluation Metrics

Following previous works (Yang et al., 2018), we adopt Hamming Loss (**HL**) and Micro-F1 (**F1**) scores as our main evaluation metrics. To further

| Dataset | $\mathcal{N}$ | $\mathcal{L}$ | $\overline{\mathcal{L}}$ | $\overline{\mathcal{N}}$ |
|---|---|---|---|---|
| **AAPD** | 55,840 | 54 | 2.4 | 163.4 |
| **RCV1-V2** | 804,414 | 103 | 3.2 | 123.9 |

Table 2: Statistics of datasets. $\mathcal{N}$ and $\mathcal{L}$ denote the total number of texts and labels. $\overline{\mathcal{L}}$ is the average number of labels associated with the text. $\overline{\mathcal{N}}$ means the average length of all texts.

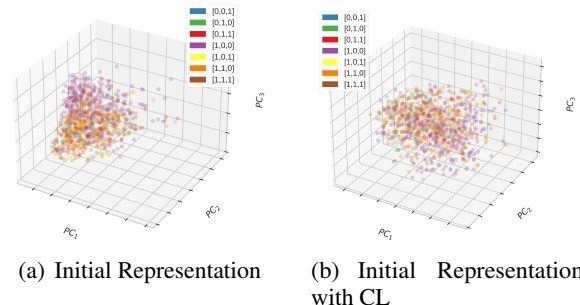

(a) Initial Representation     (b) Initial Representation with CL

Figure 3: Text representations of test set generated by initial models in 3D PCA. Where in $[h,m,l]$, $h = 1$ denotes that the current sample contains high-frequency label, while $m = 1$ and $l = 1$ denote the occurrence of middle and low-frequency label respectively.

evaluate our method, we also take Micro-Precision (**Pre**) and Micro-Recall (**Rec**) as auxiliary analysis metrics.

### 4.3 Baseline Methods

We compare our method with two sets of competitive MLTC methods including: text representation learning and label correlation exploring methods.

The first set of methods involves (1) CNN (Kim, 2014) a fine-grained text representation learning method that utilizes multiple convolutional kernels to extract text representations. (2) BERT (Devlin et al., 2019), a coarse-grained text representation learning method which takes the [CLS] token as a global classification representation. (3) LEAM (Wang et al., 2018) proposes joint embedding of text and labels to obtain text representation. (4) LSAN (Xiao et al., 2019) proposes an adaptive attention fusion strategy and classifies each document by building label-specific representation.

The second set of methods involves (5) SGM (Yang et al., 2018) applies the sequence generation model to transform the MLTC problem into sequence generation problem, and proposes global embedding mechanism to capture label correlations. (6) Seq2set (Yang et al., 2019) adds a Set decoder on the basis of the SGM and exploit of the disorder of Set to reduce the impact of incorrect label sorting. (7) LACO (Zhang et al., 2021) utilizes multi-task framework with two auxiliary label correlation learning tasks and one MLTC task. (8) CL-BERT+$k$NN (Su et al., 2022) proposes a $k$NN mechanism according to the relevant labels between samples and designs a multi-label contrastive learning objective to enhance the effect of $k$NN mechanism on MLTC.

## 5 Results and Analysis

### 5.1 Main Results

Table 3 displays the results of all compared methods on two datasets. It is noted that CL-MIL outperforms all baselines considerably on major

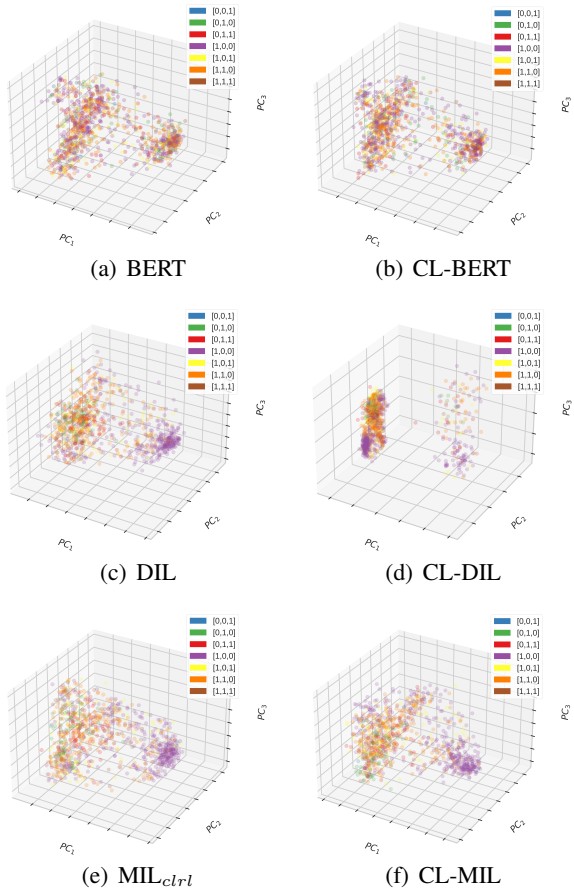

(a) BERT     (b) CL-BERT

(c) DIL     (d) CL-DIL

(e) $\text{MIL}_{clrl}$     (f) CL-MIL

Figure 4: Text representations of test set generated by fine-tuned models in 3D PCA. We present different scenarios of applying CL to the backbone models, and illustrate the specific conditions for performance improvement of applying CL.

evaluation metrics such as **Rec** (72.5/87.2), **HL** (0.0208/0.0068), and **F1** (75.8/88.9).

For text representation learning, when compared to coarse-grained learning methods like BERT, which uses the [CLS] token for global representa-

| Model | AAPD dataset | | | | RCV1-V2 dataset | | | |
|---|---|---|---|---|---|---|---|---|
| | HL(-) | Pre(+) | Rec(+) | F1(+) | HL(-) | Pre(+) | Rec(+) | F1(+) |
| **CNN** (Kurata et al., 2016) | 0.0256 | **84.9** | 54.5 | 66.4 | 0.0089 | **92.2** | 79.8 | 85.5 |
| **BERT** (Devlin et al., 2019) | 0.0230 | 76.7 | 69.8 | 73.1 | 0.0078 | 89.1 | 85.6 | 87.3 |
| **LEAM**[†] (Wang et al., 2018) | 0.0261 | 76.5 | 59.6 | 67.0 | 0.0090 | 87.1 | 84.1 | 85.6 |
| **LSAN**[†] (Xiao et al., 2019) | 0.0242 | 77.7 | 64.6 | 70.6 | 0.0075 | 91.3 | 84.1 | 87.5 |
| **LACO**[†] (Zhang et al., 2021) | 0.0213 | 80.2 | 69.6 | 74.5 | 0.0072 | 90.8 | 85.6 | 88.1 |
| **SGM**[†] (Yang et al., 2018) | 0.0251 | 74.6 | 65.9 | 69.9 | 0.0081 | 88.7 | 85.0 | 86.9 |
| **Seq2set**[†] (Yang et al., 2019) | 0.0247 | 73.9 | 67.4 | 70.5 | 0.0073 | 90.0 | 85.8 | 87.9 |
| **LACO**$_{plcp}^{†}$(Zhang et al., 2021) | 0.0212 | 79.5 | 70.8 | 74.9 | 0.0070 | 90.8 | 86.2 | 88.4 |
| **LACO**$_{clcp}^{†}$ (Zhang et al., 2021) | 0.0215 | 78.9 | 70.8 | 74.7 | 0.0070 | 90.6 | 86.4 | 88.5 |
| **CL-BERT**$_{kNN}^{†}$ (Su et al., 2022) | 0.0216 | - | - | 75.1 | 0.0071 | - | - | 88.3 |
| **CL-MIL** (Ours) | **0.0208** | 79.3 | **72.5** | **75.8** | **0.0068** | 90.6 | **87.2** | **88.9** |

Table 3: Predictive performance of each comparing algorithm on two datasets. Hamming Loss (HL), Micro Precision (Pre), Recall (Rec), F1-Score (F1) are used as evaluation metrics. The (-) represents the lower score the better performance, and the (+) is the opposite. Models with † denote for its results are quoted from previous papers.

tion in classification, CL-MIL improves the **F1** by increasing the **Pre**. Compared to fine-grained learning methods like CNN and LACO, which focus on more detailed information, CL-MIL increases the **Rec** and improves the **F1**. This suggests a need for an appropriate balance of different granularities information to improve the overall ability of learning text-labels correlations, and more specific analysis are in Section 5.3.

Among label correlations learning methods, LACO$_{plcp}$, LACO$_{clcp}$, and CL-BERT$_{kNN}$ provide more significant improvements than other baseline methods. Comparing to these three strong methods, CL-MIL reduces the **HL** by 1.9%~3.7% / 2.9%~4.2%, and improves the **F1** by 0.9%~1.5% / 0.5%~0.7% on two datasets.

| Model | AAPD | | | |
|---|---|---|---|---|
| | HL(-) | Pre(+) | Rec(+) | F1(+) |
| **BERT** | 0.0230 | 76.7 | 69.8 | 73.1 |
| **+CL** | 0.0298 | 63.8 | 78.1 | 70.1 |
| **DIL** (Ours) | 0.0214 | 78.3 | 72.2 | 75.1 |
| **+CL** | 0.0217 | 77.6 | 72.5 | 74.9 |
| **MIL**$_{clrl}$ (Ours) | 0.0211 | 78.8 | 72.4 | 75.5 |
| **+CL** | 0.0208 | 79.3 | 72.5 | 75.8 |

Table 4: Ablation over BERT, DIL and MIL$_{clrl}$ with or without CL on AAPD dataset.

## 5.2 The Impact of Contrastive Learning

For contrastive learning (CL), we utilize BERT (with joint embedding), DIL and MIL$_{clrl}$ as the backbone models to verify the effect of CL and visualize the text representations of test set generated by these models in 3 dimension PCA. As

shown in Figure 3, in initial stage, the representations after CL distribute more uniformly in geometry space. Besides, samples that only correlated high-frequency labels are preliminary pushed away from the central space of samples of low-frequency middle-frequency labels. The refined representations will be more suitable for MIL module.

Table 4 and Figure 4 demonstrate the performance changes of applying CL to backbone models with varying degrees of label information: BERT (only embeddings), DIL (Interactive Learning), and MIL$_{clrl}$ (label relations). For BERT, which learns no label information, applying CL may be counterproductive and greatly hinder performance. Similarly, CL offers no improvement to DIL, which only learns text-label correlations. However, implementing CL in MIL$_{clrl}$, which learns constraint label relations, can effectively improve the performance on main evaluation metrics. We believe the fundamental reason for this is that the training target of CL for MLTC is to push non-neighbor samples apart based on label co-occurrence, which aligns precisely with the objective of CLRL. This observation suggests that CL for MLTC has high-order relevance to CLRL.

In conclusion, CL brings a more uniform and task-adapted initial representation, and further improves the auxiliary effectiveness of CLRL by exploiting the contrastive relations among low co-occurrence frequency labels.

## 5.3 Analysis of Dual-grained Interactive Learning

We conduct an ablation study to validate the reliability and effect of DIL. Table 5 shows the compari-

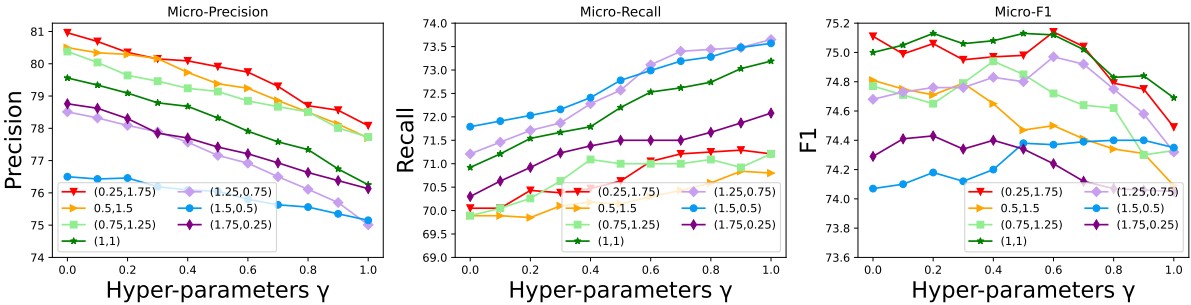

Figure 5: Hyper-parameters analysis of DIL on AAPD dataset

| Model | AAPD dataset | | | | RCV1-V2 dataset | | | |
|---|---|---|---|---|---|---|---|---|
| | HL(-) | Pre(+) | Rec(+) | F1(+) | HL(-) | Pre(+) | Rec(+) | F1(+) |
| CIL | 0.0229 | 76.8 | 70.2 | 73.5 | 0.0075 | 88.5 | 87.3 | 87.9 |
| FIL | 0.0298 | 76.2 | 66.1 | 71.1 | 0.0074 | 89.3 | 86.6 | 87.9 |
| DIL | 0.0214 | 78.3 | 72.2 | 75.1 | 0.0073 | 89.1 | 87.3 | 88.2 |

Table 5: Analysis of coarse-grained interactive learning (CIL), fine-grained interactive learning (FIL) and DIL on AAPD and RCV1-V2 datasets.

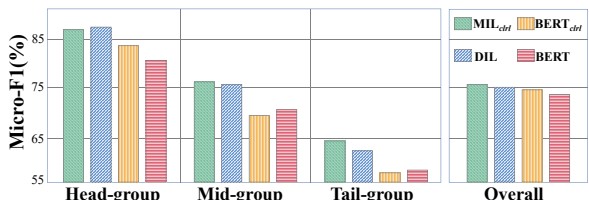

Figure 6: Micro-F1 performance on four label frequency distributions.

son of DIL with coarse-grained interactive learning (CIL) and fine-grained interactive learning (FIL) respectively. The result reveals that it is an appropriate integration of CIL and FIL that DIL achieves effective performance rather than two dispersed modules.

We also conduct a hyper-parameters analysis of DIL to further illustrate the effect of different granularities on classification performance. Figure 5 shows the influence of training loss weights $\alpha$ and $\beta$ on **Pre**, **Rec** and **F1** score under different proportion of predictions $\gamma$. The curves prove that coarse-grained information is related to **Rec** while fine-grained information correlate to **Pre**. Besides, the curves also illustrate that an appropriate ratio of training loss weights determines the lower bound of classification performance, while an apt proportion of predictions determines the upper.

## 5.4 Analysis of Constraint Label Relations Learning

According to Table 3, we can notice that label correlations learning methods obtain higher **Rec** than text representation learning methods. It indicates that these label correlations learning methods improving **F1** without considering constraint relations are based on increasing **Rec** at the cost of decreasing **Pre**. A potential reason is that learning label correlations explores whether the labels share co-occurrence in current sample, which may lead to a certain degree of label bias (high co-occurrence frequency but not correlated in current sample). It will hinder the performance since the labels have been already embedded together with the text in a same space and sufficiently interacted with the text.

Differ from these methods, our $\text{MIL}_{clrl}$ takes constraint label relations into consideration, which judges the relations not only by the co-occurrence of current sample but also by the labels co-occurrence frequency. By this means, the label bias can be alleviated, and $\text{MIL}_{clrl}$ can better enhance the FIL module to obtain more considerable **F1** with both **Pre** and **Rec** improved.

To further illustrate the effectiveness of CLRL, we analyze the label frequency of the AAPD test set and divide the labels into three groups[1] according

[1]The head label group (head-group), the middle-frequency group (mid-group) and the tail label group (tail-group) according to the label distribution of training set.

to the definition shown in Table 6. As shown in Figure 6, $MIL_{clrl}$ outperforms DIL in the mid-group and tail-group, while $BERT_{clrl}$ only improves performance in the head-group compared to BERT. Additionally, when CLRL is applied to BERT without interactive text-label correlation learning, performance decreases in the mid-group and tail-group labels, demonstrating that CLRL can better explore label relations and improve low-frequency performance under the reinforcement of DIL.

| Label Groups | Definition |
|---|---|
| head-group | For any label $L_i$ in head-group, $Frequency\,(L_i) \geq 3000$. |
| mid-group | For any label $L_i$ in mid-group, $3000 > Frequency\,(L_i) \geq 1000$. |
| tail-group | For any label $L_i$ in tail-group, $1000 > Frequency\,(L_i)$. |

Table 6: Definition of label frequency groups

## 6   Conclusion

This paper proposed a multi-granularity information enhanced MLTC framework. Through extensive experiments, we demonstrated the competitiveness and stability of our method and confirmed the effectiveness of incorporating multi-granularity information to enhance representations in MLTC. Additionally, our extended analysis highlighted the complementarity of the modules in our framework and emphasizes the necessity of each module.

## Limitations

While our method has yielded promising results by utilizing BERT as the encoder of our framework, it comes with certain limitations worth considering. Firstly, the foremost limitation is the restricted input length of the BERT encoder set at 512. In extreme label scenario, it's not feasible to embed all the labels, limiting our approach to cases with a normal number of labels. In future work, we could explore a hierarchical label embedding strategy or migrate to other pre-trained language models with unlimited embedding lengths. Secondly, although our framework explores the multi-granularity information after joint embedding of text and labels, it would be engaging to investigate the effect of label embedding order. Hence, trying different label embedding orders will provide potential insights for future work and enhance the performance of MLTC tasks.

## Ethics Statement

Our method is used to address multi-label text classification. Therefore, ethical considerations of text classification models generally apply to our method. We encourage users to assess potential biases before deploying text classification models.

## Acknowledgements

This research is supported by the National Natural Science Foundation of China [62172449, 72374070, 62006251], Hunan Provincial Natural Science Foundation of China [2021JJ30870, 2022JJ3021, 2021JJ40783], Changsha Municipal Natural Science Foundation [kq2202300], Training Program for Excellent Young Innovators of Changsha [kq2107004], The science and technology innovation Program of Hunan Province [2022RC1105] and Industry-University-Research Innovation Fund of Chinese University[2021ITA01023(mxl)]. And this research was supported in part by the High Performance Computing Center of Central South University.

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

# A  Implementation Details

We implement our model based on Bert4keras and run on NVIDIA Tesla V100s. The English BERT version is base-uncased BERT with 12 layers of encoder and 768 of hidden size. On contrastive learning stage, we use a maximum input sequence length of 230 and a batch size of 32 for AAPD, and use a maximum input sequence length of 200 and a batch size of 32 for RCV1-V2. On MLTC stage, we use maximum input sequence length of 320 and a batch size of 32 for both datasets. We train all the models on both stages and both datasets up to 20 epochs with an early stop of 2 patience and take the Adam as our optimizer with a learning rate of $5 \times 10^{-5}$.

# B  The PCA Visualization Details

In Figure 3 and Figure 4, we utilize 3 dimensional PCA visualization tool (Bengfort and Bilbro, 2019) to illustrate the text representations (to be fed into MIL module) distribution in geometry space. The text representations are divided into seven categories according to the frequency distribution of their corresponding labels in the training set.