# OpenReview forum: "Towards Better Representations for Multi-Label Text Classification with Multi-granularity Information"
_EMNLP/2023/Conference — EMNLP 2023 Findings_

### Official Review · Reviewer_WQiA · 2023-07-25

**Typos Grammar Style And Presentation Improvements:** L_CIL, L_FIL, and  L_CLRL should be e…
**Soundness:** 3

**Excitement:**

3: Ambivalent: It has merits (e.g., it reports state-of-the-art results, the idea is nice), but there are key weaknesses (e.g., it describes incremental work), and it can significantly benefit from another round of revision. However, I won't object to accepting it if my co-reviewers champion it.

**Paper Topic And Main Contributions:**

The paper addresses multi-label classification with text labels by utilizing contrastive learning as the base learning method and utilizes dual-graded interactive learning and constraint label relations learning to improve the performance considering the label frequency and correlation information.

With those approaches, they achieved strong hamming loss and F1 score, showing the word/label bias issue is mitigated, helping multi-label classification with text labels.

However, F1 score improvement is mainly due to the balance between precision and recall (increased recall while losing precision.) So, it is unsure if the proposed method is really competitive since some of the existing methods could easily increase F1 with the precision/recall balancing.
Also, there are too many hyperparameters to optimize and certain model decisions (e.g., using DGCNN) are not clearly described considering their complexities.



**Reasons To Accept:**

With the proposed approach, they achieved strong hamming loss and F1 score, showing the word/label bias issue is mitigated, helping multi-label classification with text labels.


**Reasons To Reject:**

1. F1 score improvement is mainly due to the balance between precision and recall (increased recall while losing precision.) So, it is unsure if the proposed method is really competitive since some of the existing methods could easily increase F1 with the precision/recall balancing.

2. There are too many hyperparameters to optimize and certain model decisions (e.g., using DGCNN) are not clearly described considering their complexities.

**Reproducibility:**

3: Could reproduce the results with some difficulty. The settings of parameters are underspecified or subjectively determined; the training/evaluation data are not widely available.

**Reviewer Confidence:**

4: Quite sure. I tried to check the important points carefully. It's unlikely, though conceivable, that I missed something that should affect my ratings.

---

> ### Author Rebuttal · Authors · 2023-08-28
>
> Dear Reviewer,
> We would like to express our gratitude to the reviewer for the careful reading and valuable comments. And we have carefully considered each comment and made necessary revisions to address the concerns raised. For presentation suggestion, in the revised manuscript, we will provide additional explanations and clarify each equation and symbols throughout the manuscript to reduce the confusions and improve the overall readability. Here we will explain the concerns point by point.
>
> $\textbf{Question-1:}$ F1 score improvement is mainly due to the balance between precision and recall (increased recall while losing precision.) So, it is unsure if the proposed method is really competitive since some of the existing methods could easily increase F1 with the precision/recall balancing.
> $\textbf{Answer-1:}$ We agree that this balance is crucial in assessing the competitiveness of our method. However, we would like to emphasize that our method offers additional advantages beyond mere F1 score improvement. To support the competitiveness of our method, we conducted several ablation studies.
> * We demonstrated the enhancement of text representation quality achieved by our method from a novel perspective using 3D PCA visualization. This perspective allows us to evaluate whether the word/label bias issue gets sufficient alleviation, which also further reinforces the effectiveness of our approach.
> * We also evaluated the performance of our method across different label frequency groups. The results demonstrated an overall improvement among all label groups, further validating the competitiveness of our approach.
> * Our method achieved competitive performance in terms of Hamming Loss (another evaluation metric). The performance on F1 and Hamming Loss strengthens our claim of competitiveness.
>
> $\textbf{Question-2:}$ There are too many hyperparameters to optimize and certain model decisions (e.g., using DGCNN) are not clearly described considering their complexities.
> $\textbf{Answer-2:}$  To capture more comprehensive information, we use DGCNN as the core of the coarse-grained information extraction layer. In DGCNN,
> * The dilated convolution is used to expand the receptive field of the overall structure so that it can capture more distant information.
> * The gated structure is used to transmit the coarse-grained text label interactive information in multiple channels.
> * The multi-head attention structure completes the integration of the entire sequence information.
>
> In the revised manuscript, we will provide a detailed description of hyperparameters settings and a clear explanation of the DGCNN architecture and its advantages in capturing interactive information in the representation sequence. These clarifications will help readers better understand the complexities involved and provide a clear understanding of the optimization process.

---

### Official Review · Reviewer_xJF7 · 2023-08-04

**Soundness:** 4

**Excitement:**

4: Strong: This paper deepens the understanding of some phenomenon or lowers the barriers to an existing research direction.

**Paper Topic And Main Contributions:**

This paper proposes a multi-granularity information enhancement framework to improve text representations and in effect multi-label text classification (MLTC). To achieve that the authors use contrastive learning to implicitly enhance the initial representations of the texts with the label frequencies. Given a source sample containing a low-frequency label, the authors create a positive pair using dropout and negative pairs using texts that contain frequent labels and their labels do not co-occur frequently with the labels of the source sample.

Then the authors learn representations that encode fine-grained interactions using a CNN over the similarity matrix of the representations of the labels tokens and the text tokens, and coarse-grained interactions using a 15-layerDilated CNN and a Multi-head Attention over the text token representations. The fine-grained representations are passed through an MLP that performs MLTC. Similarly, the coarse-grained representations are passed through another MLP that performs MLTC.

The authors also employ Constrained Label Relations Learning. Given two labels, A and B, the authors use a Logistic Regression classifier to determine whether A is directly, indirectly, or not related to B.

All three modules are combined using multi-task learning by operating on representations extracted by encoding all the possible pairs of texts and labels using a BERT-base model, i.e., the encoder is shared and updated by each module jointly.

The authors compare with several baselines and report improved F1 as a result of increasing recall at the expense of reduced precision. They also perform ablation tests to demonstrate the effectiveness of each component.

**Reasons To Accept:**

* The idea is interesting and the proposed method seems to be effective.
* The ablations tests and the representations demonstration support the claims of the authors.
* The hyper-parameter analysis is interesting.

**Reasons To Reject:**

* The paper is not very well written. It required quite an effort to fully understand the proposed approach.

**Reproducibility:**

3: Could reproduce the results with some difficulty. The settings of parameters are underspecified or subjectively determined; the training/evaluation data are not widely available.

**Reviewer Confidence:**

4: Quite sure. I tried to check the important points carefully. It's unlikely, though conceivable, that I missed something that should affect my ratings.

**Typos Grammar Style And Presentation Improvements:**

* Lines 182, 270, 442: The authors misuse the sense of "efficiency". "Efficiency" is usually connected with the runtime, not with the classification performance.
* Equation 11: Shouldn't it be $|\mathcal{H}_A - \mathcal{H}_B|$ or am I missing something?
* There are a few typos here and there. I suggest the authors have a careful proofreading of their manuscript.

---

> ### Author Rebuttal · Authors · 2023-08-28
>
> Dear Reviewer,
>
> We thank the reviewer for your positive and constructive feedback. And we apologize for typos and grammar mistakes. They will be corrected so that the overall writing style meets EMNLP standard. We have carefully considered your concerns and suggestions and have made the necessary revisions to address the issues you raised.
>
> $\textbf{Question-1:}$ The paper is not very well written. It required quite an effort to fully understand the proposed approach.
> $\textbf{Answer-1:}$ Regarding the concern about the overall clarity and readability, we apologize for any confusion caused by the initial manuscript. We have thoroughly revised the manuscript to improve its structure and enhance the clarity of the proposed approach to make it more readable.
>
> $\textbf{Question-2:}$Lines 182, 270, 442: The authors misuse the sense of "efficiency". "Efficiency" is usually connected with the runtime, not with the classification performance.
> $\textbf{Answer-2:}$ We agree that "efficiency" is commonly associated with runtime and not with classification performance. We have rectified this error by using “effectiveness” (lines 182, 270), “effectively” (line 442) to describe the aspects of classification performance.
>
> $\textbf{Question-3:}$ Equation 11: Shouldn't it be $\left|H_A-H_B\right|$ or am I missing something?
> $\textbf{Answer-3:}$ We highly agree with your enlightening suggestion after we reviewing the Equation 11 and relevant section. In revised manuscript, we will correct the $\left|H_{A-B}\right|$ to a more appropriate form ($\left|H_A-H_B\right|$).
>
> $\textbf{Question-4:}$ There are a few typos here and there. I suggest the authors have a careful proofreading of their manuscript.
> $\textbf{Answer-4:}$ We have carefully proofread the manuscript and addressed the typos and grammatical errors to ensure its clarity and readability.

---

### Official Review · Reviewer_HWyb · 2023-08-10

**Soundness:** 3

**Excitement:**

3: Ambivalent: It has merits (e.g., it reports state-of-the-art results, the idea is nice), but there are key weaknesses (e.g., it describes incremental work), and it can significantly benefit from another round of revision. However, I won't object to accepting it if my co-reviewers champion it.

**Missing References:**

[3] Haibin Chen, Qianli Ma, Zhenxi Lin, and Jiangyue Yan. 2021. Hierarchy-aware Label Semantics Matching Network for Hierarchical Text Classification. In Proceedings of the 59th Annual Meeting of the Association for Computational Linguistics and the 11th International Joint Conference on Natural Language Processing (Volume 1: Long Papers), pages 4370–4379, Online. Association for Computational Linguistics.

**Paper Topic And Main Contributions:**

The paper focuses on the topic of multi-label text classification and proposes a multi-granularity enhancement framework, which utilizes both fine-grained and coarse-grained information between labels. The author first proposes fine-tuning the BERT classifier with a contrastive learning objective. Then, the author proposes three different loss functions: Fine-grained Interactive Learning, Coarse-grained Interactive Learning, and Constraint Label Relations Learning. These functions aim to better learn the text-label correlation information and enhance the performance on low-frequency labels. Experimental results and ablation studies on two MLTC datasets demonstrate the effectiveness of the approach and lead to a better understanding of text representation.

**Questions For The Authors:**

A: At line 306, the author mentioned about the $lowfreq$ labels, what's the threshold for the frequency of $lowfreq$ labels? Also, what's the threshold for head-group, mid-group and tail group labels as mentioned at lines 502-504?

B: In previous works [2, 3] in representation learning for multi-label text classification, researchers usually do 2d plot on the learned representation, via t-SNE mainly but PCA can also be an option. However, this paper used 3d plot to visualize the first 3 principal component (PC) values, could the author share the variance represented by each of the $PC$ ( $PC_{1}, PC_{2}, PC_{3}$)? If $Var(PC_{1}) + Var(PC_{2}) >95 $ %, then the information carried by $PC_{3}$ bring negligible information about the representation on latent space.

C: Do the author have evaluation results (i.e. F1 or Hamming loss) when either the contrastive learning objective or Constraint Label Relations Learning objective is dropped?

[2] Zihan Wang, Peiyi Wang, Lianzhe Huang, Xin Sun, and Houfeng Wang. 2022. Incorporating Hierarchy into Text Encoder: a Contrastive Learning Approach for Hierarchical Text Classification. In Proceedings of the 60th Annual Meeting of the Association for Computational Linguistics (Volume 1: Long Papers), pages 7109–7119, Dublin, Ireland. Association for Computational Linguistics.

**Reasons To Accept:**

- A comprehensive diagram detailing the model architecture.

- Clear visualizations of the learned representations, accompanied by experiments and an ablation study that support the proposed approach for enhanced learning of representations in HTC.

- Notable improvements in classification performance on both RCV-1 and AAPD datasets, which are commonly utilized for multi-label classification."

**Reasons To Reject:**

- The methodology lacks sufficient detail and poor writing styles, making it challenging for readers to follow.
  - The definition for the frequency at which labels are categorized as head-group, mid-group, and tail-group is missing. This omission hinders the reviewer's understanding on the effectiveness for improvement of labels with varying frequency.
  - It's unclear how one transitions from $\mathcal{H_{\mathrm{relations}}}$ in Eq. (11) to $\mathcal{L_{\mathrm{CLRL}}}$ in Eq. (13), via $\mathcal{L_{\mathrm{CCE}}}$ in Eq.(12).

  - Certain equations lack clarity, and there are missing definitions for specific terms. For instance, in Eq. 13, none of the three loss symbols ($\mathcal{L}$) have been previously introduced or formally defined. Based on the reviewer's understanding, $ \mathcal{L_{\mathrm{CIL}}} $ and $\mathcal{L_{\mathrm{FIL}}}$ refer to the BCE mentioned in Eq. 8; and $\mathcal{L_{\mathrm{CLRL}}}$ refer to the Eq. 12.

- The work builds on incremental research. The primary contribution to performance seems to be the Dual-granularity Interactive Learning (DIL) method, as highlighted in Table 5. This method aims to extract joint information from text-labels, utilizing both label and document semantics via multi-headed attention (Coarse-grained) and dot product (Fine-grained). However, this concept is not novel and has been previously explored in works such as [1, 3].

- Training Complexity with Multiple Objectives: The training process of the proposed model involves optimizing multiple objectives, including the contrastive loss function, Fine-grained Interactive Learning, Coarse-grained Interactive Learning, and Constraint Label Relations Learning. This multi-objective optimization can lead to increased training complexity and may present challenges in finding a balance between the different objectives. The paper should elaborate on the training process, addressing potential challenges and demonstrating how each component contributes to the final performance. (See question C)

[1] Lin Xiao, Xin Huang, Boli Chen, and Liping Jing. 2019. Label-Specific Document Representation for Multi-Label Text Classification. In Proceedings of the 2019 Conference on Empirical Methods in Natural Language Processing and the 9th International Joint Conference on Natural Language Processing (EMNLP-IJCNLP), pages 466–475, Hong Kong, China. Association for Computational Linguistics.

**Reproducibility:**

3: Could reproduce the results with some difficulty. The settings of parameters are underspecified or subjectively determined; the training/evaluation data are not widely available.

**Reviewer Confidence:**

4: Quite sure. I tried to check the important points carefully. It's unlikely, though conceivable, that I missed something that should affect my ratings.

**Typos Grammar Style And Presentation Improvements:**

- Line 126-127, 130-131, 145, 149-150, : Change the citation styles to remove parenthesis, use \citet{CITATION} instead

- Unusual upper-case within sentence: Line 183: "Fine-tuning" -> " fine-tuning"

- At line 333, it should be $k$

- The function symbol $\mathcal{H}$ is being overloaded and kept being used ($\mathcal{H_{\mathrm{word}}}, \mathcal{H_{\mathrm{label}}^{\mathrm{Norm}}}, \mathcal{H_{\mathrm{FIL}}}, \mathcal{H_{\mathrm{relations}}}$ etc.), with different super- and subscript. This make the formalism of equations hard to follow.

---

> ### Author Rebuttal · Authors · 2023-08-28
>
> Dear Reviewer,
> We would like to extend our heartfelt appreciation to the reviewer for dedicating the time to review our submission. We are grateful for the valuable feedback provided, and we have conscientiously considered each comment to make the necessary revisions and address the raised concerns. We have diligently added the missing reference, which serves as a relevant and significant contribution to the revised manuscript. Furthermore, we will address the concerns regarding typographical errors, grammar, style, and presentation to enhance the overall quality of the paper. Here, we will provide a detailed explanation addressing each concern you have raised.
>
> $\textbf{Question-1:}$ Lacks sufficient detail and poor writing styles.
> $\textbf{Answer-1:}$ We sincerely apologize for this oversight. In response, we have meticulously revised the methodology section to offer enhanced clarity and detail regarding our proposed method. Additionally, we have made improvements to the overall writing style in our revised manuscript to ensure the readability for our readers.
>
> $\textbf{Question-1.1:}$ Missing the definition for the frequency at which labels are categorized as head-group, mid-group, and tail-group.
> $\textbf{Answer-1.1:}$ For any label $L_i$ in the head-group, the frequency should be greater than or equal to 3000. For any label $L_i$ in the mid-group, the frequency should be between 3000 and 1000. And for any label $L_i$ in the tail-group, the frequency should be less than 1000. In our revised manuscript, we have now included explicit definitions for the frequency thresholds that categorize labels into head-group, mid-group, and tail-group.
>
> $\textbf{Question-1.2:}$ It's unclear how transition from $\mathcal{H}\_{relations}$ in Eq. (11) to $\mathcal{L}\_{CLRL}$ in Eq. (13), via $\mathcal{L}\_{CCE}$ in Eq. (12).
> $\textbf{Answer-1.2:}$ We obtain the representation of label relations, $\mathcal{H}\_{relations}$, by concatenating the individual representations $\mathcal{H}\_A$, $\mathcal{H}\_B$, and $\mathcal{H}\_{A-B}$. We then apply a fully connected layer with a $softmax$ activation as the label relations classifier and use Categorical Cross Entropy, denoted as $\mathcal{L}\_{CCE}$, as the corresponding loss function. In order to improve the clarity of this transition, we will provide a more comprehensive explanation in our revised manuscript.
>
> $\textbf{Question-1.3:}$ Certain equations lack clarity, and there are missing definitions for specific terms. For instance, in Eq. 13, none of the three loss symbols ($\mathcal{L}$) have been previously introduced or formally defined.
> $\textbf{Answer-1.3:}$ The loss functions for the multi-label text classifier are defined as: $\mathcal{L}\_{CIL}$ and $\mathcal{L}\_{FIL}$ utilize Binary Cross Entropy, while $\mathcal{L}\_{CLRL}$ corresponds to the loss function of the label relations classifier, employing Categorical Cross Entropy. We will introduce and formally define these three loss symbols ($\mathcal{L}$) in our revised manuscript. Additionally, we will provide clear explanations of each equation and symbol throughout the manuscript to enhance readability.
>
> $\textbf{Question-2:}$ The primary contribution to performance seems to be the Dual-granularity Interactive Learning (DIL) method, as highlighted in Table 5. This method aims to extract joint information from text-labels, utilizing both label and document semantics via multi-headed attention (Coarse-grained) and dot product (Fine-grained). However, this concept is not novel and has been previously explored in works such as [1, 3].
> $\textbf{Answer-2:}$
> * The DIL module significantly enhances the performance of our approach. However, it is worth noting that DIL's effectiveness is limited when applied to labels with mid-group and tail-group distribution. Hence, our research emphasizes not only on DIL but also incorporates additional modules to establish a comprehensive multi-granularity enhancement framework. Together, these modules alleviate the impact of label frequency, improve the performance of low-frequency labels, and enhance the quality of text representation. We firmly believe that the synergy of these components is what makes our method competitive.
> * While we acknowledge that the concept of leveraging multi-grained information has been explored in previous studies (e.g., [3]), those primarily focused on the label perspective, modeling the correlation between multi-granularity labels and text. On the contrary, our method places greater emphasis on the text itself, facilitating the learning of interactive information between multi-grained text features and labels. This primarily aims to refine text representations and capture intricate correlations between text and labels.
>
> $\textbf{Question-3:}$ Training Complexity with Multiple Objectives: The training process of the proposed model involves optimizing multiple objectives. This multi-objective optimization can lead to increased training complexity and may present challenges in finding a balance between the different objectives. The paper should elaborate on the training process, addressing potential challenges and demonstrating how each component contributes to the final performance.
> $\textbf{Answer-3:}$ In our ablation study, we thoroughly evaluate the contribution of each component to the final performance.
> * In Tab. (4), we provide a detailed analysis when the contrastive learning objective is dropped (referred to as $MIL_{clrl}$). We also investigate the scenario when the CLRL objective is dropped ($DIL+CL$) and the case when both the contrastive learning and CLRL objectives are dropped ($DIL$).
> * Fig. (5) and Tab. (5) further explore and confirm the learning balance of DIL across different granularities during the training process.
>
> In the revised manuscript, we will provide a more comprehensive description of the training process. Additionally, we will outline a detailed procedure for optimizing multiple objectives, thereby effectively addressing the challenges encountered and highlighting the impact of each objective on the overall performance.
>
> $\textbf{Question-4(A):}$ At line 306, the author mentioned about the $lowfreq$ labels, what's the threshold for the frequency of $lowfreq$ labels? Also, what's the threshold for head-group, mid-group and tail group labels as mentioned at lines 502-504?
> $\textbf{Answer-4(A):}$ We apologize for any lack of clarity in explaining the threshold for categorizing labels as $lowfreq$ at line 306. The threshold for determining the $lowfreq$ labels is based on the average frequency co-occurrence of each label, calculated using Eq.(10): $\overline{\mathcal{M}}\_i = \frac{1}{l-1}\sum\_{j=1,j\neq i}^l\mathcal{M}\_{i,j}$. The $lowfreq$ labels $Y_i^{lowfreq}$={ $y\_a,\cdot\cdot\cdot,y\_b$} are determined according to the value of $\overline{\mathcal{M}}\_i$, where $\mathcal{M}\_{i,a} , \mathcal{M}\_{i,b}\leq \overline{\mathcal{M}}\_i$.
> For the thresholds of the head-group, mid-group, and tail-group labels:
> * For any label $L_i$ in head-group, $Frequency\left(L_i\right)\geq3000$;
> * For any label $L_i$ in mid-group, $3000>Frequency\left(L_i\right)\geq1000$;
> * For any label $L_i$ in tail-group,  $1000>Frequency\left(L_i\right)$.
>
> $\textbf{Question-5(B):}$ In previous works [2, 3] in representation learning for multi-label text classification, researchers usually do 2d plot on the learned representation, via t-SNE mainly but PCA can also be an option. However, this paper used 3d plot to visualize the first 3 principal component (PC) values, could the author share the variance represented by each of the $PC (PC_1, PC_2, PC_3)$? If $Var\left(PC_1\right)+Var\left(PC_2\right)>95%$, then the information carried by $PC_3$ bring negligible information about the representation on latent space.
> $\textbf{Answer-5(B):}$ We appreciate the reviewer's suggestion to provide variance information for the first three principal components ($PC_1$, $PC_2$, and $PC_3$) in the 3d plot visualization. When we performed 3d reduction on the 768-dimensional vector representation, we found that:
> * $Var\left(PC_1\right)=0.5630604$;
> * $Var\left(PC_2\right)=0.24930695$;
> * $Var\left(PC_3\right)=0.18763264$,
>
> which proved the feasibility of using 3d visualization. On the other hand, our choice of 3d visualization serves to prove the effectiveness of our method more clearly.
>
> $\textbf{Question-6(C):}$ Do the author have evaluation results (i.e., F1 or Hamming loss) when either the contrastive learning objective or Constraint Label Relations Learning objective is dropped?
> $\textbf{Answer-6(C):}$ We apologize for the unclear description of the ablation study. We have conducted experiments to evaluate the performance in Tab. (4) under three conditions:
> 1. the contrastive learning objective is dropped ($MIL_{clrl}$);
> 2. the constraint label relations learning objective is dropped ($DIL+CL$);
> 3. both contrastive learning and constraint label relations learning objectives are dropped ($DIL$).
>
> | **Model** | **HL(-)** | **Pre(+)** | **Rec(+)** | **F1(+)** |
> | --- | --- | --- | --- | --- |
> | **BERT** | 0.0230 | 76.7 | 69.8 | 73.1 |
> | **+CL** | 0.0298 | 63.8 | 78.1 | 70.1 |
> | **DIL (Ours)** | 0.0214 | 78.3 | 72.2 | 75.1 |
> | **+CL** | 0.0217 | 77.6 | 72.5 | 74.9 |
> | **MIL$_{clrl}$ (Ours)** | 0.0211 | 78.8 | 72.4 | 75.5 |
> | **+CL** | 0.0208 | 79.3 | 72.5 | 75.8 |
>
> *Table4: Ablation over BERT, DIL and MIL$_{clrl}$ with or without CL on AAPD dataset.*

---

### Meta-Review · Area_Chair_wT6d · 2023-09-06

**Recommendation:** 3

**Metareview:**

This paper proposed an interesting idea, an effective proposed method, supporting ablation tests and representation demonstrations, and insightful hyper-parameter analysis. However, this paper has limitations in poor writing quality, unclear transitions in equations, a lack of detail in methodology, and potential concerns about the competitive advantage of the proposed method in terms of F1 score improvement.

---

### Decision · Program_Chairs · 2023-10-07

**Decision:**

Accept-Findings

**Comment:**

This paper proposed an interesting idea, an effective proposed method, supporting ablation tests and representation demonstrations, and insightful hyper-parameter analysis. However, this paper has limitations in poor writing quality, unclear transitions in equations, a lack of detail in methodology, and potential concerns about the competitive advantage of the proposed method in terms of F1 score improvement.